# Study protocol for the Multiple Symptoms Study 3: a pragmatic, randomised controlled trial of a clinic for patients with persistent (medically unexplained) physical symptoms

Cara Mooney [ORCID],[1] David Alexander White,[1] Jeremy Dawson [ORCID],[2] Vincent Deary,[3] Kate Fryer,[4] Monica Greco,[5] Michelle Horspool,[6] Aileen Neilson [ORCID],[7] Gillian Rowlands,[8] Tom Sanders,[9] Ruth E Thomas,[10] Steve Thomas,[6] Waquas Waheed,[11] Christopher D Burton [ORCID] [4]

For numbered affiliations see end of article.

**Correspondence to**
Cara Mooney;
c.d.mooney@sheffield.ac.uk

## ABSTRACT

**Introduction** Persistent physical symptoms (which cannot be adequately attributed to physical disease) affect around 1 million people (2% of adults) in the UK. They affect patients' quality of life and account for at least one third of referrals from General Practitioners (GPs) to specialists. These referrals give patients little benefit but have a real cost to health services time and diagnostic resources. The symptoms clinic has been designed to help people make sense of persistent physical symptoms (especially if medical tests have been negative) and to reduce the impact of symptoms on daily life.

**Methods and analysis** This pragmatic, multicentre, randomised controlled trial will assess the clinical and cost-effectiveness of the symptoms clinic intervention plus usual care compared with usual care alone. Patients were identified through GP searches and mail-outs and recruited by the central research team. 354 participants were recruited and individually randomised (1:1). The primary outcome is the self-reported Physical Health Questionnaire-15 at 52 weeks postrandomisation. Secondary outcome measures include the EuroQol 5 dimension 5 level and healthcare resource use. Outcome measures will also be collected at 13 and 26 weeks postrandomisation. A process evaluation will be conducted including consultation content analysis and interviews with participants and key stakeholders.

**Ethics and dissemination** Ethics approval has been obtained via Greater Manchester Central Research Ethics Committee (Reference 18/NW/0422). The results of the trial will be submitted for publication in peer-reviewed journals, presented at relevant conferences and disseminated to trial participants and patient interest groups.

**Trial registration number** ISRCTN57050216.

## STRENGTHS AND LIMITATIONS OF THIS STUDY

⇒ The symptoms clinics are delivered by specially trained GPs in a structure that would allow broader roll out if shown to be effective.
⇒ Patients with lived experience were involved in the design of the trial and will provide advice throughout delivery.
⇒ Blinding of participants was not feasible due to the nature of the intervention.
⇒ Measures are taken to reduce the impact of this including blinding outcome data collectors and trial statisticians.
⇒ The embedded process evaluation will allow us to understand how the intervention works in practice and identify the processes underlying the outcomes.

## INTRODUCTION
### Background and rationale

Persistent physical symptoms (PPS), which cannot be adequately attributed to physical disease, affect approximately 1 million adults in the UK (2% of the adult population).[1 2] Many patients with such symptoms receive repeated referral and investigation[3] which provides little benefit[4] but has real costs to health services time and diagnostic resources.[5] When patients are told that medical tests do not show a cause for their symptoms they are commonly disappointed in their interactions with clinicians.[6 7] Patients want to have those symptoms explained in acceptable ways[8 9] in order to know that their symptoms are legitimate,[6] to adapt to them and to manage them. Without an explanation for their symptoms many patients seek further healthcare use while at the same time losing confidence that it will help them. With acceptable explanation, patients may be able to move from looking for a cause, to self-managing their symptoms.[7]

PPS represents a broad category of disorders, including defined syndromes such as fibromyalgia or irritable bowel syndrome but also non-specific symptoms and combinations of symptoms from different syndromes.[10 11] The term replaces older and unhelpful terms including 'medically unexplained symptoms'.[12] Recent thinking suggests that PPS, such as chronic pain, should be regarded as disorders in their own right.[10] This fits with models of symptoms as consequences of disturbed interoception—the non-conscious sensing, interpreting and regulating the body.[13–15] Disturbed interoception, or more specifically disturbed predictive interoceptive coding,[15] provides a fundamental explanation for PPS which can then be elaborated in a number of psychological[16] and biopsychosocial[17] models for persistent symptoms. These often use frameworks such as predisposing, precipitating and persisting factors to structure explanations.

We developed a model of 'rational explanation',[18] which enables clinicians to integrate knowledge from processes such as disturbed interoception, with patients' reported experiences, to develop explanations for symptoms. These rational explanations make sense of symptoms in terms of brain and body processes and are acceptable to doctor and patient.[19 20] They leave room for psychosocial influences without placing them as the cause, and they provide opportunities to guide self-management, which has been found to be of value to patients.[21] In rational explanations, psychological factors such as heightened vigilance to symptoms or persistent worry about symptoms are presented as understandable mechanisms by which symptoms persist rather than signs that symptoms have a 'psychosomatic' cause. In contrast, previously advocated explanatory models such as somatisation are rejected by patients as too simplistic[8 9] and leave patients with PPS dissatisfied with the explanations they receive. Rational explanations based on signalling between the brain and the body also open up the possibility of using symptom management techniques which influence interoception and the autonomic nervous system including slow paced breathing.[22]

Improving PPS could have a substantial effect on health and on its impacts in terms of lost productivity and increased care needs. Physical symptoms not explained by disease account for very substantial costs[5]—between 40% and 60% of all referrals across a range of specialties,[4] estimated at £3 billion annually to the National Health Service (NHS) and £14 billion to the wider economy.[23]

The symptoms clinic is a primary care intervention, designed to explore acceptable explanations for symptoms and to reduce the impact of PPS on daily life. The Multiple Symptoms Study 3 (MSS3), randomised control trial (RCT), builds on successful preliminary studies which have shown the feasibility, and acceptability of the symptoms clinics.[24 25]

The primary aim of MSS3 is to determine the clinical and cost-effectiveness of the 'symptoms clinic' intervention for patients with persistent ('medically unexplained') physical symptoms.

## Objectives

1. Conduct a pragmatic RCT, with internal pilot, of the symptoms clinic verses usual care, in people with PPS.
2. Establish symptoms clinics for the purposes of the trial, train Extended Role General Practitioners (ER-GP) and provide them with supervision; systematically recruit patients from primary care, and ensure satisfactory trial procedures and follow-up.
3. Compare patient experience of physical symptoms and quality of life, as well as healthcare use, across 52 weeks, between participants allocated to the symptom clinic plus usual care and those allocated to usual care.
4. Understand the processes of change associated with the symptoms clinic by (A) conducting qualitative interviews with a subsample of participants (B) recording and coding key elements of the intervention and (C) interviewing stakeholders.

## METHODS AND ANALYSIS
### Trial design
MSS3 is a pragmatic, multicentre, parallel group, individually randomised controlled trial, with internal pilot. It uses a superiority framework to compare the symptoms clinic intervention plus usual care to usual care alone.

### Adaptations in response to the COVID-19 pandemic
The MSS3 RCT was originally designed and delivered as a face-to-face intervention. Prior to March 2020, enrolment appointments and delivery of the symptoms clinic took place in local GP practices or community research facilities. After a short pause due to COVID-19 restrictions the trial was redesigned to allow for remote delivery as described in this protocol. No changes were made to the content of the intervention. Sensitivity analyses will be conducted to explore differences in those receiving the intervention face to face and remotely, with a further sensitivity analysis removing those cases that were randomised immediately before the pause, for whom there was a substantial delay in the delivery of the symptoms clinic (so whose 13-week outcomes were sometimes collected before the intervention had begun; those randomised to the usual care group during the same period will also be removed for this sensitivity analysis). Qualitative interviews will explore participant and stakeholder opinion of the different delivery modalities.

### Participants
Participants were recruited in four areas: Yorkshire and the Humber, Greater Manchester, Newcastle and Gateshead, and Northwest London.

### Inclusion and exclusion criteria
Inclusion criteria:
1. Aged between 18 and 69 years (inclusive) at the time of the computer search.

2. Current physical symptoms which meet the below criteria.
   a. Clinical records suggest PPS.
   b. Records show at least two referrals for specialist opinion in the last 36 months (extended to 42 months when restarting after the first pandemic wave).
   c. Records show no evidence of any previous or current major illnesses likely to cause multiple symptoms.
   d. Doctors in the GP practice do not believe that the majority of the patient's symptoms can be currently explained by other pathology.
   e. The score on the Physical Health Questionnaire-15 (PHQ-15) is between 10 and 20 (inclusive).
3. Access to a mobile phone with video calling capability or an email address and computer with video conferencing capability.

Exclusion criteria:
1. A score of 3 on question 9 on the PHQ-9 completed at the enrolment appointment.
2. Difficulty conducting a healthcare consultation in English without either a professional or family interpreter or other assistance.
3. The GP regards inviting them to participate as inappropriate (eg, recent bereavement).
4. Severe symptom-related disability (eg, requiring help with daily personal care or severely impaired mobility).
5. Undergoing active multidisciplinary rehabilitation, Improving Access to Psychological Therapies (IAPT) programme or specialist psychological treatment including specialist pain, fatigue or other symptom clinic at the time of screening.
6. Currently pregnant or less than 6 months postnatal at the time of the screening telephone call.

A three-stage identification process was adopted: computer searching, GP record screening and postal invitation.

## Computer searching
GP practices ran a computer search to identify patients. The search strategy is listed in online supplemental materials 1.

## GP record screening
A GP at the practice screened the list produced by the computer search to exclude patients for whom invitation may be inappropriate (eg, major medical conditions not included in the search or concern about the appropriateness of invitation).

## Postal invitation
The GP practices sent invitation packs containing an invitation letter, Participant information sheet (PIS), PHQ-15 and a reply form with a prepaid return envelope. Interested patients returned the reply form and the completed PHQ-15 to the clinical trials unit. Reminder invitation packs were sent to non-responders approximately 3 weeks after the initial mailing. Respondents whose PHQ-15 was outside the eligible range were sent a letter and received no further contact.

## Recruitment and informed consent
Potentially eligible patients, based on their PHQ-15 score, were contacted by the research team to provide further information and answer questions. If the patient wished to proceed with the study, the research team completed screening checks and if appropriate, scheduled a study enrolment appointment. During the enrolment appointment, a member of the recruitment team answered any final questions, obtained informed consent (see online supplemental materials 2), confirmed eligibility and collected baseline data. Figure 1 presents the participant flow through the trial.

## Usual care
All participants continued to receive usual care from their registered general practice. While the intervention was delivered by specially trained GPs, this was done outside of their usual clinical practices and thus they had no contact with participants outside of the trial. The GP providing usual care was informed of their patients' participation and group allocation. GPs of patients allocated to the symptoms clinic also received a letter from the symptoms clinic doctor (copied to the participant) after the first and last appointments which outlined the formulation and any planned actions.

## Randomisation and blinding
Following consent and baseline data collection, participants were individually randomised (1:1) to the symptoms clinic plus usual care or usual care alone, using a computer generated pseudorandom list, stratified by study centre with random permuted blocks of varying sizes. Allocation was concealed using a centralised web-based randomisation system.

The participant was then randomised and informed of their allocation. If assignment was to the intervention, the first symptoms clinic appointment was scheduled. If assignment was to usual care alone participants were reminded to use their usual healthcare services (eg, usual care GP) as required.

Due to the nature of the intervention, it is not possible to blind participants to their allocation. For practical reasons such as coordinating symptoms clinic appointments and ER-GP supervision some members of the research team are not blinded, including the trial manager and chief investigator (CI).

Members of the trial steering committee (TSC), study statisticians, health economists and those collecting outcome data are blinded to treatment allocation while the trial is ongoing.

## ER-GP recruitment, training and supervision
Seven ER-GPs were recruited and trained to deliver the symptoms clinic. Two withdrew because of competing demands, one after seeing fewer than 5 patients and one before seeing any.

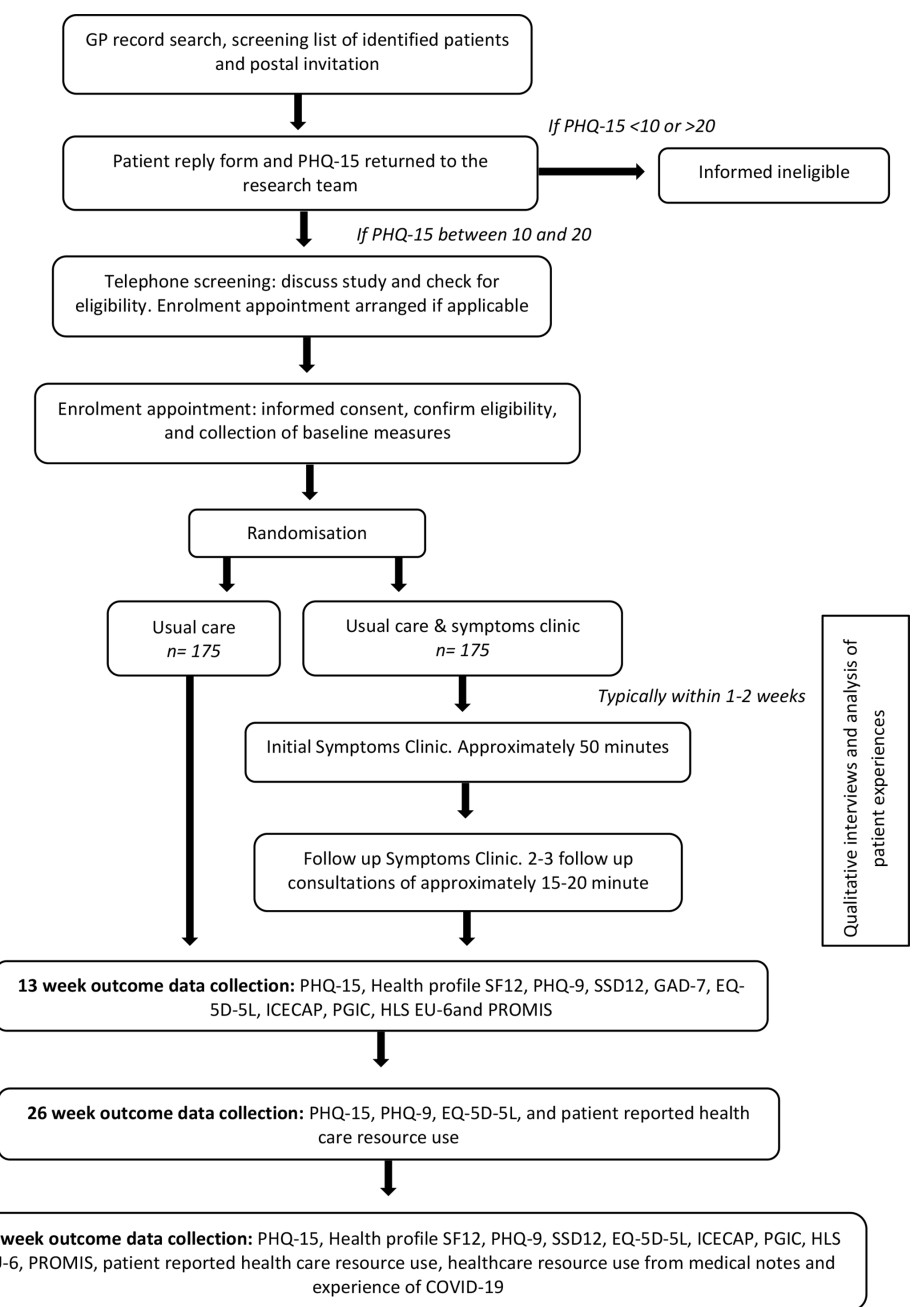

**Figure 1** Participant flow diagram.

Training comprised a mixture of small group sessions (both didactic and interactive), protected time to conduct and reflect on symptom clinic consultation techniques in practice and one-to-one or small group supervision. It involved 13 half-day sessions. Sessions 1–4 were two full days of training. Sessions 5–7 and 9–11 comprised protected time to see patients of the GPs own practice using newly learnt skills and reflection on this. Sessions 8 and 12–13 were training sessions focusing on consolidating skills and knowledge. In sessions 9–11 each ER-GP recorded a set of three consultations for review, quality assessment and constructive feedback by a panel comprising the CI and two other investigators.

During the study, ER-GPs received supervision with one of the investigators approximately every 1–2 months.

Supervision included review of consultation content and encouraged reflective learning and consolidation of existing knowledge and skills and learning of new knowledge and skills.

### The symptoms clinic

The symptoms clinic intervention is a sequence of medical consultations which aim to elicit a detailed clinical history, ensure that the patient's experience is fully heard and validated, to offer rational explanations for symptoms and to assist the patient to develop ways of managing their symptoms. The treatment model can be summarised under four headings: Recognition, Explanation, Action and Learning (see online supplemental materials 3 for further details).

Consultations before March 2020 were delivered face to face. Subsequently consultations took place via video consultation or telephone. The symptoms clinic consists of up to four consultations; an initial long consultation (approximately 50 min) followed by up to three medium length consultations (15–20 min) approximately every 2 weeks. Clinicians had flexibility to increase the gaps between sessions if required.

### Fidelity of the symptoms clinic intervention

All symptoms clinic consultations were audiorecorded. A random sample of approximately one-third are transcribed for quality assurance and process assessment and the remainder are archived for quality assurance purposes. It was ensured that a sample from each ER-GP was selected.

Fidelity is assessed from consultation transcripts or recordings against standards developed in the preliminary studies. The protocol originally proposed that this would include the proportion of consultation time spent on different components and the number and type of explanations. These proved difficult to operationalise and a simpler approach was adopted in which a framework of items in the intervention was used as a template and for each consultation the presence of each item was indicated and evidenced by using an extract or quote from the transcript. A traffic light system was used where clearly present was marked green, possibly present marked amber and absent marked red.

### Symptoms clinic attendance

Participants received appointment reminder text messages the day before each symptoms clinic appointment, which were personalised to include their name, ER-GP name and appointment details. Attendance was monitored using the study database where re-arranged and missed appointments were recorded.

### Outcomes

The primary outcome is the PHQ-15[26] at 52 weeks postrandomisation. The PHQ-15 consists of 15 items for which patients are asked to report symptom severity over the past 4 weeks on a scale of 0 (not bothered at all), 1 (bothered a little) or 2 (bothered a lot). It has excellent internal reliability ($\alpha$=0.80) and good convergent validity with other measures of functionality, symptom severity and disability days.[26]

The secondary outcomes are:
► Quality of Life measured using the EuroQol 5 dimension 5 level version (EQ-5D-5L).[27] In addition we will collect data for Short-Form Six Dimension (SF-6D)[28] derived from the 12-Item Short Form Health Survey (SF-12) and ICECAP-A[29 30] to compare their performance in this study population.
► Symptoms of depression and anxiety using the PHQ-9[31] and Generalised Anxiety Disorder-7.[32 33]
► Healthcare resource (HCRU) use over the 52-week period using both self-report (a bespoke resource use questionnaire capturing healthcare use, in primary and secondary care as well as NHS and private services) and medical case note review of GP records.
► Patient-reported Global Indicator of Change.
► Ability to Participate in Social Roles and Activities - Patient Reported Outcome Measurement Information System (PROMIS)[34]

In addition, we will measure:
► Somatic Symptoms Disorder-B criteria scale-12.[35]
► European Health Literacy Survey-6.[36]

We are also collecting data on whether the participants have experienced symptoms of COVID-19.

### Data collection and management

Self-report measures are collected by questionnaire at the enrolment appointment and by post at 13, 26 and 52 weeks postrandomisation. Non-responders are followed up.

Researchers collecting and handling outcome measures will be blinded to participant allocation. The extraction of healthcare use data from medical records will be completed after all other measures have been collected from the participant and in a prespecifed order of extraction to minimise the risk of the outcome data collector becoming unblinded through exposure to correspondence in the notes before other data are collected.

If, at any stage, the outcome data collector know (or suspect) they have been unblinded, this will be recorded.

Data will be recorded in paper case report forms or online at the time of each participant contact. All forms use anonymised participant ID codes to protect confidentiality. Data are entered into the trial unit's web-based data management system (Prospect), by authorised members of the research team. All data are collected and retained in accordance with the Data Protection Act 2018, the General Data Protection Regulation and trial unit standard operating procedures (SOPs).

### Participant retention

Participant retention is promoted through communication from the research team which clearly explains the importance of completing outcome data regardless of study arm. This message is reinforced at enrolment and all follow-up points. The questionnaire cover letter explains the importance of every returned questionnaire and participants are offered a £10 voucher on completion of the 52-week questionnaires.

### Intervention completion and withdrawal

Intervention completion is defined as having an initial consultation and at least one follow-up consultation.

Participants may withdraw either from the intervention only or the trial and this is documented. If the participant withdraws from the trial, no further data will be collected.

### Patient and public involvement

People with lived experience of PPS were involved in the design and development of MSS3. Patient participation was incorporated in the delivery of the project through

representation in the trial management group (TMG) and TSC.

## Sample size

In the pilot trial, we observed an average 3.2 point clinically important change in the intervention group from baseline to 13 weeks, compared with a 1.4 point change in the control group. We have thus powered the trial on a between group difference of two points on the PHQ-15 (equivalent to a clinically important three point change from baseline).

We have based calculations of effect size on a pooled SD of 5; this is larger than that seen in our preliminary studies owing to their restricted eligibility range and more in keeping with observational studies. This results in a standardised effect size of 0.4, which is similar to that seen in two small European studies of extended GP consultations for broadly comparable patients.[37 38]

### Calculation of sample size

Allowing 25% lost to follow-up, and a further pragmatic 6% inflation to allow for minor treatment centre imbalances or differences, a sample of 188 patients per arm has 90% power (alpha=0.05) to detect this effect. The initial recruitment target was thus 376 participants. In October 2021, this was reduced in discussion with the funder, to 350 because lost to follow-up at 52 weeks postrandomisation was 18% rather than the anticipated 25%.

## Data analysis

The primary outcome will be analysed using a partially nested heteroscedastic mixed-effects model to account for clustering by clinic GP. Secondary outcomes will be analysed in a similar manner within a generalised linear modelling framework using appropriate link functions for the outcomes' distributions. Models will adjust for sex, age, whether the intervention was delivered in person or online, and baseline values of outcomes. A repeated measures analysis on PHQ-15 at all four measurement points will be conducted as a further secondary analysis using a multilevel growth curve model with time as a quadratic term, and a treatment-time interaction included in the model.

Intention-to-treat analysis will be used for the primary analysis of all outcomes, with complier average causal effect analysis as a secondary analysis. The primary outcome will be analysed using observed data with no imputation for missing data, but we will assess the amount and patterns of missing data and test the sensitivity of estimates of treatments effects using an appropriate imputation strategy such as multiple imputation by chained equation. We will explore potential modification of the treatment effect by including treatment-by-subgroup interactions in models. All treatment effect estimates will be presented with 95% CIs in forest plots.

A single main analysis will be performed at the end of the trial when follow-up is complete. Interim analyses will be performed if requested by the data monitoring and ethics committee (DMEC) and the trials unit SOPs will be adhered to maintain the integrity of the trial.

## Process evaluation

The process evaluation comprises three nested observational studies, including consultation content analysis and interviews with participants and key stakeholders.

### Consultation content analysis

A sample of approximately 30% of consultations are transcribed. These will be used to examine the intervention content using the classification of consultation content, explanations and response to explanation which we have developed from the preliminary studies.[19 20 39] We will use this data to conduct exploratory analysis relating to explanation type, content and negotiation to patient outcomes in order to develop better understanding of the mechanisms by which the intervention affects outcomes.

### Participant and stakeholder interviews

To explore processes of change within participants, semistructured interviews are conducted with a purposive sample of participants at different stages of the intervention. Only participants who did not have their consultations transcribed were invited for interview, and an even distribution was sought between patients with pain and fatigue type symptoms, and working and non-working patients. Interviews will be transcribed and analysed thematically, recognising that there are likely to be changes in intrapersonal understanding and interpretation (for which an interpretive phenomenological approach is likely to be valuable) and interpersonal or social understanding and interaction. Particular attention will be paid to patients' views on what aspects of the Symptoms Clinic were particularly valuable to them and how these translated into perceived changes in thoughts, behaviours and symptoms.

Stakeholder interviews will examine acceptability of the clinic concept and processes, skills learnt and knowledge transferred, value for GPs and perceived value to patients.

### Relationship between process evaluation and intervention delivery

Medical Research Council (MRC) guidance on process evaluation highlights the importance of considering the relationship between process evaluation and intervention delivery[40] including whether the process evaluation is allowed to inform the intervention or the two are independent of each other. Information was permitted to flow from the process evaluation to the intervention during the first 3 months of symptoms clinic delivery. These can be considered as the time of professional learning curves for both the ER-GPs and the supervising investigators. During this time early lessons can be learnt and shared.

## Health economics

We will conduct a cost-effectiveness analysis (CEA) of the symptoms clinic plus usual care compared with usual care alone from the primary perspective of the UK NHS and personal social services. This will be based on

HCRU (including primary and secondary care contacts such as GP consultations, diagnostic tests and investigations, physical and mental health specialist referrals, and prescription psychotropic and pain-related medications) and outcome data collected during the trial. It will take the format of a within-trial CEA and use a cost-utility framework to estimate cost per quality-adjusted life-year (QALY) gained.

The effects of the intervention will be estimated as gain in QALYs at 52 weeks using health-related quality of life data collected at baseline, 13, 26 and 52 weeks and the area under the curve method. Published UK tariffs will be used to convert these data to quality of life weights.

We will measure preference-based health-related quality of life using the EQ-5D-5L and the SF-6D. We will also use the newer capability well-being ICECAP-A measure to examine their relative responsiveness to change in this patient population.

A self-reported healthcare resource use questionnaire will be administered at 26 and 52 weeks postrandomisation to estimate healthcare resource use costs.

Data from GP electronic records at 52 weeks postrandomisation will be collected, where available and used for cross validation with self-reported data. Data from GP records will be extracted onto a standardised data collection form.

Use of healthcare resources will be valued and the associated costs estimated by assigning unit costs from standard published UK sources (including personal social service research unit unit costs, NHS reference costs, British National Formulary). Costs related to intervention delivery will be estimated using trial records, taking into account:

▶ Face-to-face/video consultation clinic time.
▶ Clinic-related administration.
▶ Clinician training,
▶ Clinical supervision.

The CEA will be performed on an intention to treat basis (for participants with complete data on resource use and health utilities across all follow-up time points). The results of the analysis will be reported as incremental costs, effects and incremental cost-effectiveness ratios in terms of the incremental cost per QALY gained.

Generalised linear regression analyses will be used to estimate the differences (and associated 95% CIs) in per patient mean total costs and differences in mean total QALYs comparing the symptoms clinic intervention plus usual care compared with usual care alone, adjusting for baseline differences in cost, utility and other patient characteristics (eg, age, gender, PHQ-15 score). Uncertainty will be explored by conducting a range of one-way and multiway deterministic sensitivity analyses (or probabilistic sensitivity analysis if more appropriate) to test the robustness of the base case results including assuming a broader cost perspective (eg, including private healthcare costs), evaluating the effect of missing values (comparing results based on complete cases and those estimated using multiple imputed values) and potential bias due to high-cost patients (removing these expensive participants from the analysis). Cost per QALY data will also be presented in the form of cost-effectiveness acceptability curves to show the probability that the intervention is cost effective for different values of willingness to pay per additional QALY.

### Study within a trial
A study within a trial will evaluate the impact of a pen and a brief PIS on levels of participant recruitment, using a factorial embedded RCT. Patients were randomised to: (1) A pen with the trial logo printed on, in addition to the standard invitation materials; (2) A pen with the trial logo printed on, in addition to a brief PIS, and the standard invitation materials; (3) A brief PIS and the standard invitation materials or (4) The standard invitation materials alone.

### Ethics and dissemination
#### Safety
Adverse events (AEs) may be identified during participant consultations or from self-report measures. We will only collect AEs defined as 'expected' for this trial which include (A) significant exacerbation of mental distress defined as a PHQ-9 score of 20 or more and/or a score of 2 or 3 on question 9 (suicidality item), representing at least a 1 point score change (ie, a change from 2 to 3 from their previous measure), (B) self-harm and (C) emerging serious mental illness or substance use disorder identified after randomisation. All AEs which meet the definition of serious AE (SAE) will be collected and assessed for relatedness to the intervention. Related SAEs will be reported to the Sponsor and the Research Ethics Committee.

#### Governance
Sheffield Clinical Trials Research Unit on behalf of the Sponsor (NHS Sheffield Clinical Commissioning Group, 722 Prince of Wales Road, Darnall, Sheffield S9 4EU) coordinates the trial. The CI, project coapplicants, members of the data management team, sponsor, trial manager and other representatives form the TMG, who oversee the operation of the trial and through which amendments will be communicated. The TSC, comprised two clinicians, a statistician, Health Economist and PPI representative, provides independent oversight. The independent DMEC comprised two clinicians and a statistician reviews the trial data and advises the TSC on issues of patient safety and trial continuation.

#### Ethics approval
This trial was approved by Greater Manchester Central Research Ethics Committee, reference 18/NW/0422), on 25 June 2018. The committee will be notified of any amendments to the trial protocol as appropriate.

#### Dissemination
We will publish the study's findings in peer-reviewed academic journals and present at local, national and international conferences where possible. We will publish

a short summary of the results on the MSS3 website that can be accessed by all trial participants as well as relevant interest groups.

**Author affiliations**
[1]Clinical Trials Research Unit, School of Health and Related Research, University of Sheffield, Sheffield, UK
[2]Sheffield University Management School, University of Sheffield, Sheffield, UK
[3]Psychology, Northumbria University, Newcastle, UK
[4]Academic Unit of Primary Medical Care, The University of Sheffield, Sheffield, UK
[5]Department of Sociology, Goldsmiths University of London, London, UK
[6]NHS Sheffield Clinical Commissioning Group, Sheffield, UK
[7]Usher Institute, University of Edinburgh Division of Medical and Radiological Sciences, Edinburgh, UK
[8]Population Health Sciences Institute, Newcastle University Institute for Health and Society, Newcastle upon Tyne, UK
[9]Social Work, Education and Community Wellbeing, Northumbria University, Newcastle upon Tyne, UK
[10]Centre for Healthcare Randomised Trials (CHaRT) Health Services Research Unit, University of Aberdeen, Aberdeen, UK
[11]Centre for Primary Care, University of Manchester, Manchester, UK

**Contributors** CDB, CM, DaW, JD, VD, MG, MH, AN, GR, TS, WW and RET cowrote the original trial protocol. CM lead the development of the protocol for publication and wrote the initial draft, KF, CDB, TS and MG developed the process evaluation section. All authors contributed to reviewing and revising the draft versions prior to submission.

**Competing interests** None declared.

**Patient and public involvement** Patients and/or the public were involved in the design, or conduct, or reporting, or dissemination plans of this research. Refer to the Methods section for further details.

**Patient consent for publication** Not applicable.

**Provenance and peer review** Not commissioned; externally peer reviewed.

**ORCID iDs**
Cara Mooney http://orcid.org/0000-0002-3086-7348
Jeremy Dawson http://orcid.org/0000-0002-9365-8586
Aileen Neilson http://orcid.org/0000-0003-3758-0566
Christopher D Burton http://orcid.org/0000-0003-0233-2431

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
