## [Reviewer comments · BMJ Open]

ARTICLE DETAILS

TITLE (PROVISIONAL)	Study protocol for the Multiple Symptoms Study 3:A pragmatic, randomised controlled trial of a clinic for patients with persistent (medically unexplained) physical symptoms
AUTHORS	Mooney, Cara; White, David; Dawson, Jeremy; Deary, Vincent; Fryer, Kate; Greco, Monica; Horspool, Michelle; Neilson, Aileen; Rowlands, Gillian; Sanders, Tom; Thomas, Ruth; Thomas, Steve; Waheed, Waquas; Burton, Christopher

VERSION 1 – REVIEW

REVIEWER	Peter Lucassen Radboud University Medical Center, Department of Primary and Community Care
REVIEW RETURNED	02-Aug-2022

GENERAL COMMENTS	This is a study protocol for an ongoing study about the effectiveness of the ‘Symptoms clinic intervention plus usual care’ versus usual care alone for patients with persistent physical symptoms, an important problem in primary care. Participants will be followed for 52 weeks. Explaining PPSs to patients is a widely recognized problem in primary care, so the study is very relevant and timely. The description of the study is thorough and in accordance with the CONSORT statement. Important topics in an RCT such as randomization, allocation of randomization order, blinding of trial personnel are adequate. The internal validity is high. Patient involvement ensures appropriateness of the intervention towards participants. I’m impressed by the work that has been done, its thoroughness. It must be a hell of a job to organize this very relevant trial. I have only some minor comments: • Page 4 line 27: under (a) it is stated that there are qualitative interviews with participants and under (c) there will be interviews with participants and stakeholders. In the process evaluation on page 10 there are only interviews with participants and stakeholders. I assume that (a) is an error.• Page 7 line36 states that 1/3 of the consultations will be transcribed. Is this a random sample of all consultations?• Page 9 lines 29-33: in the description of clinically relevant difference it is stated that 3.2 points difference from baseline is relevant. Is this on a 10-point scale or....? Please add the scale width.• Page 10 line 44: a purposive sample of participants will be interviewed. What are the criteria for purposive sampling? Age? Sex? Etc.• Concerning dissemination (page 14): I’m curious if the authors see the intervention as something that has to be applied by specialized
---

	general practitioners, or that is should be disseminated to education programs for general practitioner trainees. I would recommend the latter possibility. In conclusion, very relevant, very well described trial.
--	--

REVIEWER	Lisa M. McAndrew US Department of Veterans Affairs, Veterans Affairs War Related illness and Injury Study Center
REVIEW RETURNED	11-Aug-2022

GENERAL COMMENTS	Thank you for the opportunity to review “A pragmatic, randomized controlled trial of a clinic for patients with persistent (medically unexplained) physical symptoms” This protocol is for an important pragmatic, multi-centre, randomized controlled trial that will assess the clinical and cost-effectiveness of the Symptoms Clinic intervention plus usual care compared to usual care alone. Persistent physical symptoms (PPS) are a prevalent and disabling problem. The investigators are leaders in this area and present a nice overview of the problem including references older terminology. “This fits with models of symptoms as consequences of disturbed interoception – the non-conscious sensing, interpreting and regulating the body.” This is a very simplified description. I understand that models of PPS are complex and beyond the scope of this paper, but I would encourage the investigators to provide a succinct description of the complex interaction of biological, psychological and social factors that are predisposing, precepting, and perpetuating. The model of rational explanation is exciting and well-described in the introduction. Methods The usual care arm could be better described. Did they get sent to one of the trained providers? Did providers know the usual care were in the study? For the intervention arm, the providers were already their usual care providers? How did it work that they then switched them to receive the symptom clinic intervention? More information about the delivery of the interventions within usual care clinics is needed. The measures could be better described? There were a lot of acronyms which made it difficult to read. What is HCRU? CRFs? CTRU? Overall, additional information about the methods would be helpful.
---

VERSION 1 – AUTHOR RESPONSE

Reviewer 1

Comment	Response
Page 4 line 27: under (a) it is stated that there are qualitative interviews with participants and under (c) there will be interviews with participants and stakeholders. In the process evaluation on page 10 there are only interviews with participants and	We thank the reviewer for this query and have corrected this in the manuscript.

stakeholders. I assume that (a) is an error.	
Page 7 line36 states that 1/3 of the consultations will be transcribed. Is this a random sample of all consultations?	Yes, a random sample of consultations was selected, ensuring that consultations from each of the ER-GPs were included in the selection. We have updated the manuscript to reflect this.
Page 9 lines 29-33: in the description of clinically relevant difference it is stated that 3.2 points difference from baseline is relevant. Is this on a 10-point scale or....? Please add the scale width.	In the pilot trial we observed a clinically important average 3.2 point change in PHQ15 (range 0-30, baseline mean value 15.8).
Page 10 line 44: a purposive sample of participants will be interviewed. What are the criteria for purposive sampling? Age? Sex? Etc.	We thank the reviewer for this point and we have clarified this by including the following details: Only participants who did not have their consultations transcribed were invited for interview, and an even distribution was sought between patients with pain and fatigue type symptoms, and working and non-working patients.
Concerning dissemination (page 14): I'm curious if the authors see the intervention as something that has to be applied by specialized general practitioners, or that is should be disseminated to education programs for general practitioner trainees. I would recommend the latter possibility.	We welcome the reviewer's curiosity and assure him that we will be taking our results (positive and negative) into education programmes. We just feel it's a bit premature to say that just now. Similarly, while the trial involves specialised GPs, we are already thinking about what can be translated into routine training (but again perhaps too early to say...)

Reviewer 2

Comment	Response
"This fits with models of symptoms as consequences of disturbed interoception – the non-conscious sensing, interpreting and regulating the body." This is a very simplified description. I understand that models of PPS are complex and beyond the scope of this paper, but I would encourage the investigators to provide a succinct description of the complex interaction of biological, psychological and social factors that are predisposing, precepting, and perpetuating.	We agree that this was a little too brief, but are very conscious of both scope and word count. We have included the following with reference to a number of models. "Disturbed interoception, or more specifically disturbed predictive interoceptive coding , provides a fundamental explanation for PPS which can then be elaborated in a number of psychological and biopsychosocial models for persistent symptoms. These often use frameworks such as predisposing, precipitating and persisting factors to structure explanations. " While we agree they have value, we did not

	explicitly use these in this study.
The usual care arm could be better described. Did they get sent to one of the trained providers? Did providers know the usual care were in the study?	We have added a specific section about Usual Care. We have clarified that patients receiving usual care did not see one of our trained providers and that their usual care GP was informed of the involvement in the trial and their allocation.
For the intervention arm, the providers were already their usual care providers? How did it work that they then switched them to receive the symptom clinic intervention? More information about the delivery of the interventions within usual care clinics is needed.	See response to the above comment, hopefully this point has now also been addressed.
The measures could be better described?	We have rearranged the way these are presented so it is clear what they represent. We have not provided further detail about each of the measures because of space limitations in this protocol paper. We will however ensure that measures are appropriately described in subsequent results papers.
There were a lot of acronyms which made it difficult to read. What is HCRU? CRFs? CTRU?	We have revised the language to use simpler terms (rather than acronyms) to increase readability for a more general audience.
Overall, additional information about the methods would be helpful.	We thank the reviewer for this comment, but are very conscious of the word count. We are preparing further papers which will describe the methods in further detail including the process evaluation and the Statistical Analysis Plan will also be made publicly available.

VERSION 2 – REVIEW

REVIEWER	Peter Lucassen Radboud University Medical Center, Department of Primary and Community Care
REVIEW RETURNED	08-Sep-2022
GENERAL COMMENTS	I thank the authors for their answers.